# The Addis Declaration on Immunization: Assessing the Effectiveness and Efficiency of Immunization Service Delivery Systems in Africa as of the End of 2023

**DOI:** 10.3390/vaccines13010013

**Published:** 2024-12-27

**Authors:** Franck Mboussou, Charles Shey Wiysonge, Bridget Farham, Ado Bwaka, Sarah Wanyoike, Amos Petu, Sidy Ndiaye, Andre Bita Fouda, Johnson Muluh Ticha, Adidja Amani, Regis Obiang, Magaran Monzon Bagayoko, Benido Impouma

**Affiliations:** World Health Organization, Regional Office for Africa, Brazzaville P.O. Box 06, Congo

**Keywords:** Addis Declaration, immunization, universal access, Africa

## Abstract

**Background/Objectives**: The Addis Declaration on Immunization (ADI) is a historic pledge aiming at increasing political will to achieve universal access to immunization services and includes ten commitments to shape the future of immunization in Africa. **Methods**: To analyze African countries’ performance in achieving the fourth ADI commitment, a cross-sectional retrospective study was conducted including the 54 African Member States of the World Health Organization (WHO) out of 55 African Union (AU) Member States. The fourth ADI commitment aims at increasing the effectiveness and efficiency of immunization delivery systems and has four performance indicators. **Results**: The median percentage of districts with less than 10% of dropout rate between the first dose of diphtheria–tetanus–pertussis-containing vaccine (DTP1) and the third dose (DTP3) was 86.5%, ranging from 22% to 100%. Thirty-four countries (63%) recorded 80% or above of districts with less than 10% dropout rate between DTP1 and DTP3. Eleven countries (20.3%) and ten countries (18.5%) sustained 90% or above coverage for DTP3 and first dose of measles-containing vaccine (MCV1), respectively, in the past three years (2021–2023). Four countries (7.4%) had 44.5 skilled health workers per 10,000 people. Out of the 54 WHO Member States, 7 achieved at least three of the four indicators of the fourth ADI commitment (13%). **Conclusions**: It is critical, as a follow up to this study, to document best practices from the seven countries that achieved the fourth ADI commitment. Additionally, a deeper analysis of factors associated with achieving the ADI commitments is required.

## 1. Introduction

The World Health Organization (WHO) established the Expanded Program on Immunization (EPI) in 1974 to ensure that all children in all countries have access to life-saving vaccines [1]. Since then, immunization coverage has increased significantly. From 1980 to 2016, the WHO and United Nations Children’s Fund (UNICEF) Estimates of National Immunization Coverage (WUENIC) show increases from 12% to 82% for the Bacille Calmette-Guérin vaccine (BCG), from 14% to 83% for the first dose of diphtheria–tetanus–pertussis-containing vaccine (DPT1), and from 9% to 81% for the first dose of measles-containing vaccine (MCV1). Despite this significant progress, immunization coverage was suboptimal in most African countries as of 2016. One in five African children had still not received life-saving vaccines, leading to persistent outbreaks of vaccine-preventable diseases (VPD), pointing to gaps in immunization coverage and disease surveillance [2].

Recognizing the need to invest more in routine immunization to continue improving the health of African populations, in February 2016, the African Union (AU)’s Ministers of Health, Finances, Education, Social Affairs, and Local Governments adopted the Declaration of Addis on universal access to immunization as a cornerstone for health and development in Africa, known as the Addis Declaration on Immunization (ADI) [3]. The ADI is made up of 10 commitments and was officially endorsed by Heads of State on 31 January 2017 during the 28th African Union Heads of State Summit [2]. This endorsement represents a reaffirmation of Africa’s commitment to reaching all children with life-saving vaccines and keeping universal access to immunization at the forefront of efforts to achieve ambitious immunization goals and reduce child mortality.

The ADI is a historic pledge aiming at increasing political will to achieve universal access to immunization services and includes ten commitments to shape the future of immunization in Africa (Table 1).

The 10 ADI commitments outline concrete ways for countries to improve their immunization programs. To ensure successful implementation of the ADI, the African Union Member States agreed to establish an accountability framework comprising indicators and targets for monitoring countries’ progress [4].

The African Union Commission (AUC) and the World Health Organization (WHO), through its regional office for Africa (WHO AFRO) and for the Eastern Mediterranean (WHO EMRO), were assigned the mission of overseeing the implementation and monitoring of the ADI’s accountability framework and providing updates on progress made toward achieving the set targets [4]. In 2019, the WHO AFRO and EMRO published the first progress report on ADI implementation [5]. This showed that more than half of African countries (29 of 54) made significant progress toward achieving equitable coverage with the administration of the third dose of diphtheria–tetanus–pertussis-containing vaccine (DTP3) between 2015 and 2018 [5]. However, in terms of equity, there were still persistent barriers in vaccine and healthcare delivery systems, especially in the poorest, most vulnerable, and most marginalized communities [5]. Regarding domestic funding, only 19 out of 54 African countries (35%) increased their contributions to immunization programs [5]. From then onward, the COVID-19 pandemic substantially disrupted routine immunization services in Africa, leading to a precipitous decline in childhood vaccination rates compared to pre-pandemic levels and resulting in a greater number of un- and under-immunized children [6].

Following the lifting of the public health emergency of international concern (PHEIC) status for the COVID-19 pandemic by the WHO on 5 May 2023 [7], assessing the achievement of the ADI commitments, starting with the fourth one, is critical. The fourth commitment was selected in this study as it is related to the effectiveness and efficiency of immunization service delivery systems. Given constrained budgets and competing priorities, increasing the efficiency of health service delivery is seen as a way to free up resources for increasing the scale and quality of services [8]. This commitment provides a platform for assessing the ability of routine immunization to use the health workforce and investments in the immunization programs to sustain high-level coverage for all antigens and reduce dropout rates.

This paper summarizes progress made by African countries in achieving the indicators related to the fourth ADI commitment as of the end of 2023.

## 2. Materials and Methods

### 2.1. Presentation of the Fourth ADI Commitment

This study assesses the achievement of the fourth ADI commitment. Four indicators were defined for the fourth ADI commitment as per the accountability framework: (i) the proportion of countries with over 80% of districts that recorded less than 10% dropout rate between DTP1 and DTP3, (ii) the proportion of countries with sustained DTP3 coverage of 90% or above in the past three years, (iii) the proportion of countries with sustained MCV1 coverage of 90% or above in the past three years, and (iv) the proportion of countries with at least 80% of districts or administrative units with a minimum threshold of 44.5 skilled health workers per 10,000 people. Given that data on skilled health workers density by district were not available, the proportion of countries with a minimum threshold of 44.5 skilled health workers per 10,000 people at the national level was used as a proxy.

### 2.2. Study Design

A cross-sectional retrospective study analyzing the African countries’ performance in achieving the fourth ADI commitment was conducted.

### 2.3. Inclusion and Exclusion Criteria

Fifty-four African countries that are Member States of the WHO were included [9]. The Sahrawi Arab Democratic Republic, a country that is a Member State of the African Union but not a WHO Member State and which does not report data on immunization coverage to the WHO and UNICEF through Joint Report Forms (JRF), was not included.

### 2.4. Data Sources and Measurement

Table 2 shows the criteria of achievement and the source of data for the four indicators of the fourth ADI commitment.

### 2.5. Data Analysis

The achievement of each indicator of the fourth ADI commitment was assessed using the criteria specified in Table 2.

Using WUENIC data, the status of sustained coverage was assigned to each country for DTP3 and MCV1; countries that recorded 90% coverage for each antigen each year from 2021 to 2023 were considered those that sustained immunization coverage.

For each indicator, the percentage of countries achieving the indicator was computed by dividing the number of countries that achieved the indicator by the total number of countries, i.e., 54.

The performance in achieving the fourth ADI commitment was computed for each country by dividing the number of indicators achieved by the total number of indicators, i.e., 4.

The commitment was considered as achieved for countries with 75% or above performance, corresponding to at least 3 out of 4 indicators achieved.

Subsequently, the proportion of countries that have achieved the fourth ADI commitment was computed by dividing the number of countries that achieved the fourth ADI commitment by 54.

All data were analyzed using R version 4.2.1 [15] for statistical analysis and ESRI 2017 ArcGIS Pro 2.1.0 [16] for mapping.

## 3. Results

### 3.1. Indicator 1: Proportion of Countries with More than 80% of Districts Recording Less than 10% Dropout Rate Between DTP1 and DTP3

Figure 1 presents the distribution of the percentage of districts with less than 10% dropout rate between DTP1 and DPT3 in 2023 by country in Africa.

The median percentage of districts with less than 10% dropout rate between DTP1 and DTP3 was 86.5%, ranging from 22% in Botswana to 100% in Sao Tome and Principe, Sierra Leone, Rwanda, Libya, Lesotho, Egypt, and Djibouti.

Thirty-four countries out of fifty-four (63%) achieved the first indicator of the fourth ADI commitment, i.e., they recorded 80% or more districts with less than 10% dropout rate between DTP1 and DTP3.

At the national level, the median dropout rate was 5% [range: 0%; 20%]. Forty-five countries (83%) recorded less than 10% dropout rate at the national level against nine countries with 10% or above (17%) (Angola, Benin, Central African Republic, Chad, Democratic Republic of Congo, Guinea, Liberia, Mozambique, and Somalia).

### 3.2. Indicator 2: Proportion of Countries with Sustained Coverage of at Least 90% for DPT3 in the Past Three Years (2021, 2022, and 2023)

Figure 2 presents the distribution of DTP3 coverage in 2021, 2022, and 2023 by country in Africa.

In 2023, 19 countries out of 54 recorded at least 90% DTP3 coverage (35%), including 11 (20.3%) that sustained at least 90% DTP3 coverage in the past three years (Burkina Faso, Cabo Verde, Egypt, Eritrea, Ghana, Kenya, Mauritius, Morocco, Seychelles, Sierra Leone, and Tunisia).

### 3.3. Indicator 3: Proportion of Countries with Sustained Coverage of at Least 90% for MCV1 in the Past Three Years (2021, 2022, and 2023)

Figure 3 presents the distribution of MCV1 coverage in 2021, 2022, and 2023 by country in Africa.

In 2023, 20 countries out of 54 recorded at least 90% MCV1 coverage (37%), including 10 (18.5%) that sustained 90% or above MCV1 coverage in the past three years (Botswana, Cabo Verde, Egypt, Eritrea, Ghana, Morocco, Seychelles, Tunisia, Uganda, and Zambia).

### 3.4. Indicator 4: Proportion of Countries with a Minimum Threshold of 44.5 Skilled Health Workers per 10,000 People at the National Level in 2022

The median number of skilled health workers per 10,000 people at the national level was 12.5, ranging from 1.4 to 110.9; the mode was 8.0.

Four countries out of 54 (7.4%) surpassed 44.5 skilled health workers per 10,000 people: Eswatini (44.7/10,000), Cabo Verde (60.4/10,000), Libya (88.9/10,000), and Seychelles (110.9/10,000).

### 3.5. Achievement of the Commitment

Only 7 countries out of 54 achieved at least three of the four indictors to measure the fourth ADI commitment (13%). While Cabo Verde and Seychelles met all four indicators, Egypt, Eritrea, Ghana, Morrocco, and Tunisia each met three indicators.

Figure 4 presents the geographical distribution of the proportion of the four indicators of the fourth ADI commitment achieved in 2023 by country in Africa.

Table 3 presents a heatmap of the achievement of each of the four indicators of the ADI commitment as of the end of 2023.

## 4. Discussion

In January 2017, African Heads of State endorsed a historical pledge aiming at ensuring universal access to immunization across the continent [3]—the ADI. This high-level political commitment represents a major move to increasing and sustaining domestic investments and funding for routine immunization, but will only be beneficial for African children and families if translated into concrete actions. To achieve this, the AUC, in collaboration with the WHO and other immunization partners, developed a roadmap for the implementation of the ADI, which includes an accountability framework [4]. This framework set indicators to monitor progress in each of the 10 ADI commitments. This paper assesses progress, as of the end of 2023, of the implementation of the ADI in Africa through the fourth commitment, as per the accountability framework, while African countries are recovering from three years of the active phase of the COVID-19 pandemic.

The first indicator of the fourth ADI commitment monitors the dropout rate at the district level using dropout between DTP1 and DTP3 as a tracer. Each African country is expected to achieve a dropout rate of less than 10% between DTP1 and DTP3 in at least 80% of districts. The dropout rate is one of the determinants of immunization coverage and program performance, continuity, and follow-up [17]. According to the WHO, a dropout rate of over 10% is unfavorable and indicates underutilization of immunization services [18]. In this study, 63% of African countries recorded a dropout rate of less than 10% in over 80% of districts, versus 83% of countries recording a dropout rate of less than 10% at the national level. Low national dropout rates can conceal large subnational inequities across countries. Kirkby et al. [19] found an average of 7.3% DTP1–DTP3 dropout rate in 2018 in districts in Ethiopia, ranging from 0% to 57%. In Kenya, a study that evaluated the impact of text messaging and sticker reminders to reduce dropouts from the vaccination program in 2014 found that having a caregiver with below secondary education and residing more than 5 km away from the health facility were associated with higher odds of dropping out [18,19]; caregivers who received reminder text messages were less likely to drop out [19]. Robust defaulter tracking systems and strategies are critical to reducing missed opportunities in order to address inequalities in immunization, especially in underserved areas [18].

Large inequalities in the subnational dropout rate tend to be correlated with low DTP1 and DTP3 coverage at the national level [19]. The second and third indicators of the fourth commitment are related to the proportion of countries with sustained DPT3 and MCV1 coverages of 90% or above in the past three years. In 2023, these indicators were achieved by 11 countries (20%) for DTP3 coverage and 10 countries (18%) for MCV1. This means that most countries in Africa did not sustain DTP1 and MCV1 coverages as of the end of 2023. Several authors have reported a drop in DTP3 and MCV1 coverage during the COVID-19 pandemic period [20,21,22]. LaFond et al. [22] identified four direct drivers of routine immunization coverage improvement, including (i) engaging more with community-centered health workers, (ii) strengthening the collaboration between health districts, local governments, and community groups to plan and execute immunization services, raise awareness, and define strategies to reach underserved areas, (iii) carrying out regular immunization data reviews to inform operational decisions, and (iv) tailoring immunization services to community needs and conditions. Addressing these drivers implies implementing the “reaching every district” strategy [23], developed and introduced in 2002 by the WHO, the United Nations Children’s Fund (UNICEF), and other partners to improve immunization systems in areas with low coverage [24].

The link between health worker density and immunization services is well documented [21]. In recognition of this linkage, one of the four indicators on the fourth ADI commitment monitors the proportion of countries with a minimum threshold of 44.5 skilled health workers per 10,000 people at the national level. In this study, only 4 countries out of the 54 African countries achieved this indicator, including 1 high-income country (Seychelles), 1 upper-middle-income country (Libya), and 2 lower-middle-income countries (Cabo Verde and Eswatini). The shortage of health workers in most African countries sounds as a call to continuously strengthen health systems to achieve results in specific health programs, including immunization [21]. To this end, it is critical to invest more in human resources for health, including community health workers. Indeed, it is only by securing a sufficient, equitably distributed, adequately supported, and well-performing health workforce that any health goals and targets set by national governments or the international community can be met [25].

In this study, only 7 of the 54 African countries assessed achieved at least 75% of the four indicators of the fourth ADI commitment: Egypt, Eritrea, Ghana, Ghana, Morocco, Cabo Verde, Seychelles, and Eswatini. This means that most countries in Africa still need to put in additional efforts to achieve the fourth ADI commitment. Among the countries that achieved the fourth commitment, five are non-eligible for Gavi support, one is in the accelerated transition out of Gavi support (Ghana), and only one is in the initial self-financing phase [26]. This highlights the need to increase governments’ contributions to financing routine immunization programs as well as developing innovative funding solutions at the national level. None of the countries that have achieved the fourth ADI indicator are among the 12 countries on high or very high alert as per the fragile status index [27].

This study shows that a formal platform for regular monitoring of all ADI commitments is required, as well as better support by all stakeholders to countries’ efforts aiming at translating political commitments into concrete actions.

### Limitations

The WUENIC estimates were used to assess two of the four indicators of the fourth ADI commitment. The WHO and UNICEF annually review data on immunization coverage to estimate national coverage with routine service delivery of selected vaccines. The estimates are based on government reports submitted to the WHO and UNICEF and are supplemented by survey results from the published and grey literature and may be different from administrative and official coverage and sometimes not supported by national authorities for varying reasons. An analysis using administrative or official data on immunization coverage could lead to different results in some countries.

The assessment of the indicators related to the percentage of districts with a dropout rate of less than 10% between DTP1 and DTP3 was performed using data reported by national authorities to the WHO and UNICEF in the annual Joint Reporting Forms on Immunization. These reports are based on administrative data, which may be biased by inaccurate data on children vaccinated and/or on target populations.

The analysis of the indicator related to the health workforce used data from the WHO national workforce accounts data portal. The latest available data were for the year 2022 and were used for an assessment related to the year 2023. This may be misleading in case of significant changes in health workforce density in some countries.

The findings of this report should, therefore, be interpreted with these limitations.

## 5. Conclusions

The endorsement of the ADI by Heads of State raised the hope that universal immunization would be achieved in Africa with political support at the highest level. Six years later, the road toward universal access to immunization still seems very long. The results of this study have provided evidence that most African countries are still struggling to sustain childhood immunization coverage above set targets and to lessen subnational inequalities. The suboptimal performance of immunization programs in most countries suggests the urgent need to enhance the application of the “reaching every district” strategy to strengthen routine immunization in health districts. However, 7 countries in Africa out of 54 achieved the fourth ADI commitment in 2023 as an integrated part of strong and sustainable primary health care systems. It is critical, as a follow-up to this study, to document best practices from these seven countries that contributed to achieving the fourth ADI commitment. Additionally, a deeper analysis of factors associated with achieving the ADI commitments is required.

The AUC, with partners’ support, is urged to give a new impetus to the ADI through renewed commitment by Heads of State and a strengthened accountability framework to put most African countries on track toward achieving the ADI commitments and the 2030 Immunization Agenda targets.

## Figures and Tables

**Figure 1 vaccines-13-00013-f001:**
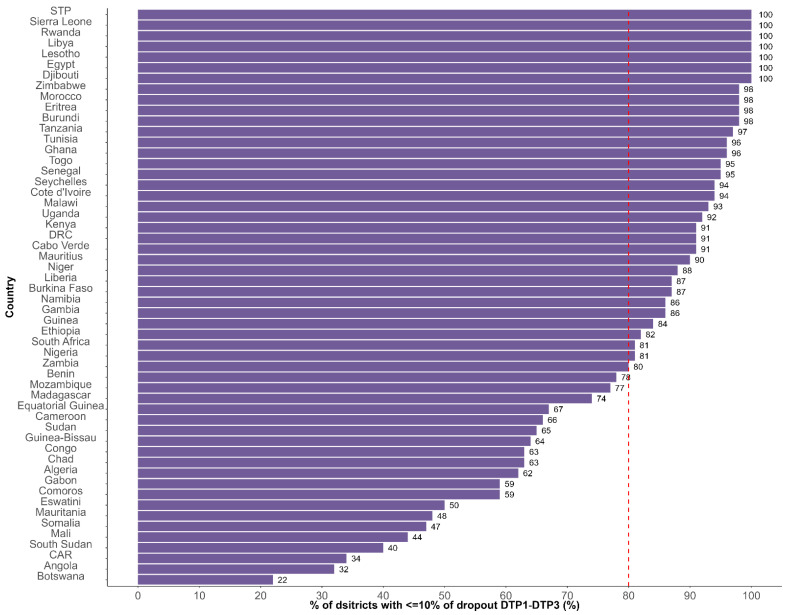
Distribution of the percentage of districts with less than 10% dropout rate between DTP1 and DTP3 in 2023 in African countries. DRC: Democratic Republic of Congo; STP: Sao Tome and Principe; CA: Central African Republic.

**Figure 2 vaccines-13-00013-f002:**
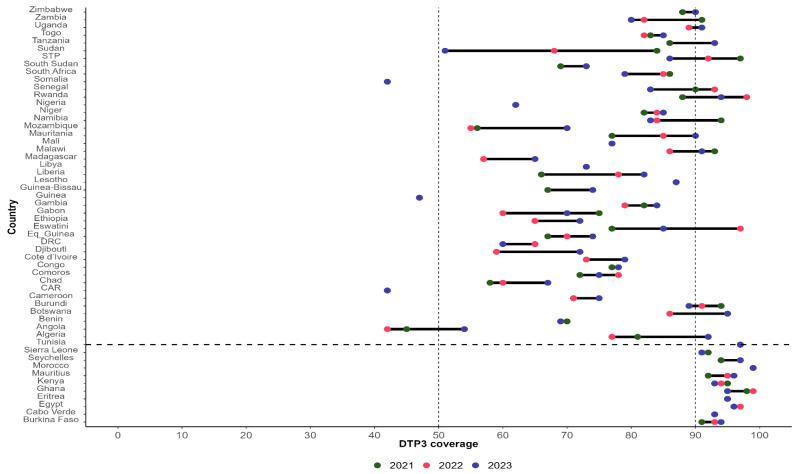
DTP3 coverage in 2021, 2022, and 2023 by country in Africa (source 2023 WUENIC revision). DRC: Democratic Republic of Congo; STP: Sao Tome and Principe; CA: Central African Republic; Eq_Guinea: Equatorial Guinea.

**Figure 3 vaccines-13-00013-f003:**
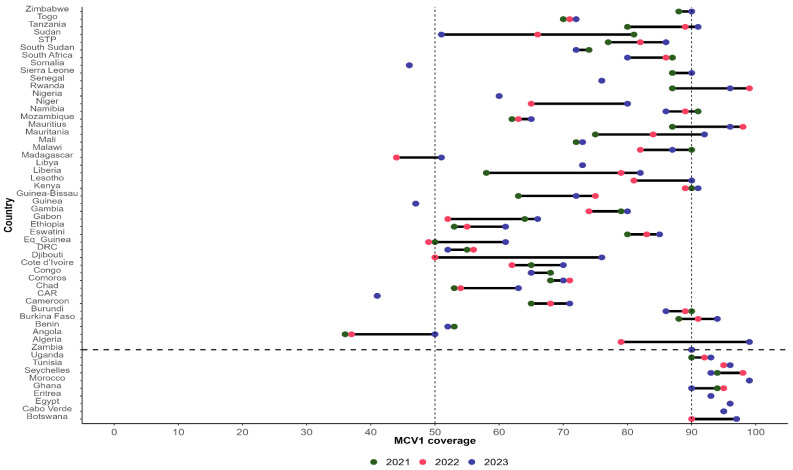
MCV1 coverage in 2021, 2022, and 2023 by country in Africa (source 2023 WUENIC revision). DRC: Democratic Republic of Congo; STP: Sao Tome and Principe; CA: Central African Republic; Eq_Guinea: Equatorial Guinea.

**Figure 4 vaccines-13-00013-f004:**
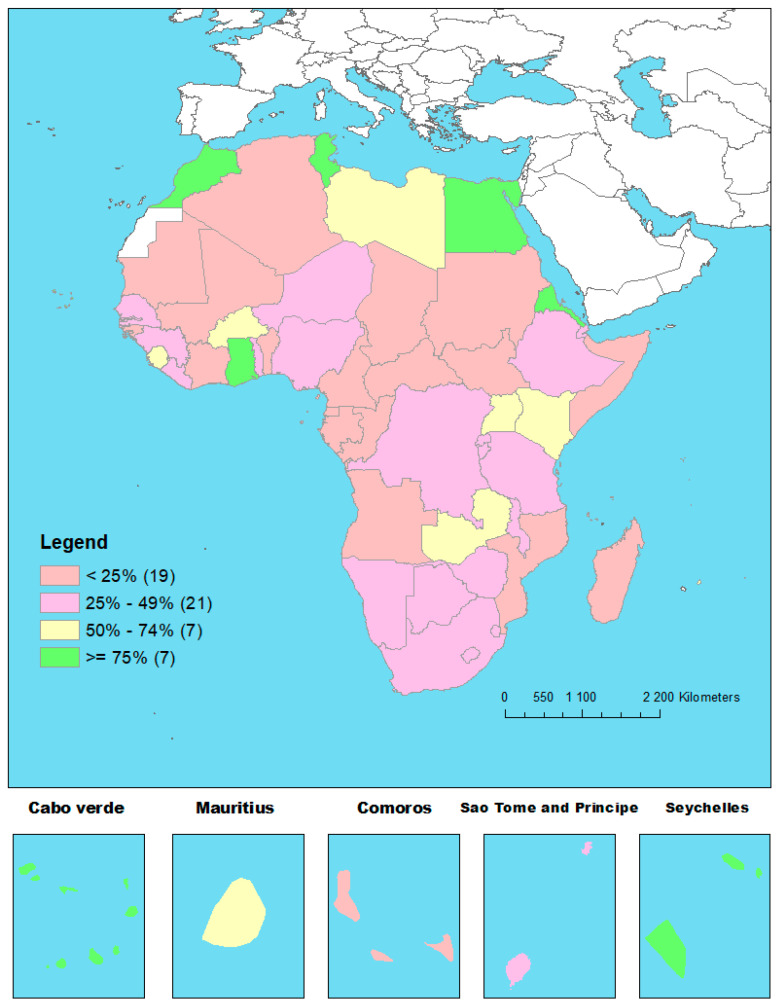
Geographical distribution of the proportion of the four indicators of the fourth ADI commitment achieved in 2023 by country in Africa.

**Table 1 vaccines-13-00013-t001:** Ten Commitments of the Addis Declaration on Immunization [3].

Commitment Number	Commitment Description
01	Keep universal access to immunization at the forefront of efforts to reduce child mortality.
02	Increase and sustain domestic investments and funding allocations for immunization.
03	Address the persistent barriers in vaccine and healthcare delivery systems, especially in the poorest, vulnerable, and most marginalized communities.
04	Increase the effectiveness and efficiency of immunization delivery systems as an integrated part of strong and sustainable primary health care systems.
05	Attain and maintain high-quality surveillance for targeted vaccine-preventable diseases.
06	Monitor progress toward achieving the goals of the global and regional immunization plans.
07	Ensure polio legacy transition plans are in place.
08	Develop a capacitated African research sector to enhance immunization implementation and uptake.
09	Promote and invest in regional capacity for the development and production of vaccines.
10	Build broad political will for universal access to life-saving vaccines.

**Table 2 vaccines-13-00013-t002:** Indicators and criteria of achievement of the fourth Addis Declaration on Immunization commitment.

ADI Indicator	Country Level Indicator	Country Target	Source of Data
1. Proportion of countries with over 80% of districts that recorded less than 10% dropout rate between DTP1 and DTP3	% of districts with a dropout rate of less than 10% between DTP1 and DTP3	≥80%	JRF * [10], accessible on WIISE Mart ** [11]
2. Proportion of countries with sustained coverage of 90% or above for DPT3 in the past three years	DTP3 coverage in the past three years	≥90% in each of the past three years	WUENIC *** [12] accessible on the WHO Immunization Data Portal [13]
3. Proportion of countries with sustained coverage of at least 90% for MCV1 in the past three years.	MCV1 coverage in the past 3 years	≥90% in each of the past three years	WUENIC
4. Proportion of countries with a minimum threshold of 44.5 skilled health workers per 10,000 people at the national level	Skilled health workers per 10,000 people at the national level	≥44.5/10,000	NHWA **** [14]

* JRF: WHO/UNICEF Joint Report Form. ** WIISE Mart: WHO Immunization Data Warehouse. *** WUENIC: 2023 revision of the WHO and UNICEF Estimates of National Immunization Coverage. **** NHWA: WHO National Health Workforce Accounts Data Portal (latest available data are from 2022).

**Table 3 vaccines-13-00013-t003:** Heatmap on the achievement of each of the four indicators of the ADI commitment as of the end of 2023.

Country	Indicator 1 Achieved?	Indicator 2 Achieved?	Indicator 3 Achieved?	Indicator 4 Achieved?	Total Indicators Achieved	% Indicators Achieved
Algeria	No	No	No	No	0	0
Angola	No	No	No	No	0	0
Benin	No	No	No	No	0	0
Botswana	No	No	Yes	No	1	25
Burkina Faso	Yes	Yes	No	No	2	50
Burundi	Yes	No	No	No	1	25
Cabo Verde	Yes	Yes	Yes	Yes	4	100
Cameroon	No	No	No	No	0	0
Central African Republic	No	No	No	No	0	0
Chad	No	No	No	No	0	0
Comoros	No	No	No	No	0	0
Congo	No	No	No	No	0	0
Cote d’Ivoire	Yes	No	No	No	1	25
Democratic Republic of Congo	Yes	No	No	No	1	25
Djibouti	Yes	No	No	No	1	25
Egypt	Yes	Yes	Yes	No	3	75
Equatorial Guinea	No	No	No	No	0	0
Eritrea	Yes	Yes	Yes	No	3	75
Eswatini	No	No	No	Yes	1	25
Ethiopia	Yes	No	No	No	1	25
Gabon	No	No	No	No	0	0
Gambia	Yes	No	No	No	1	25
Ghana	Yes	Yes	Yes	No	3	75
Guinea	Yes	No	No	No	1	25
Guinea-Bissau	No	No	No	No	0	0
Kenya	Yes	Yes	No	No	2	50
Lesotho	Yes	No	No	No	1	25
Liberia	Yes	No	No	No	1	25
Libya	Yes	No	No	Yes	2	50
Madagascar	No	No	No	No	0	0
Malawi	Yes	No	No	No	1	25
Mali	No	No	No	No	0	0
Mauritania	No	No	No	No	0	0
Mauritius	Yes	Yes	No	No	2	50
Morocco	Yes	Yes	Yes	No	3	75
Mozambique	No	No	No	No	0	0
Namibia	Yes	No	No	No	1	25
Niger	Yes	No	No	No	1	25
Nigeria	Yes	No	No	No	1	25
Rwanda	Yes	No	No	No	1	25
Sao Tome and Principe	Yes	No	No	No	1	25
Senegal	Yes	No	No	No	1	25
Seychelles	Yes	Yes	Yes	Yes	4	100
Sierra Leone	Yes	Yes	No	No	2	50
Somalia	No	No	No	No	0	0
South Africa	Yes	No	No	No	1	25
South Sudan	No	No	No	No	0	0
Sudan	No	No	No	No	0	0
Togo	Yes	No	No	No	1	25
Tunisia	Yes	Yes	Yes	No	3	75
Uganda	Yes	No	Yes	No	2	50
Tanzania	Yes	No	No	No	1	25
Zambia	Yes	No	Yes	No	2	50
Zimbabwe	Yes	No	No	No	1	25
Total countries that achieved the indicator	34	11	10	4		
% countries that achieved the indicator	63	20	19	7		
Legend						
	Achieved					
	Not achieved					

## Data Availability

The data presented in this study are available from the corresponding author upon request.

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
