# Peer review of "The Addis Declaration on Immunization: Assessing the Effectiveness and Efficiency of Immunization Service Delivery Systems in Africa as of the End of 2023"

_vaccines, 2024, doi:10.3390/vaccines13010013_

Round 1

Reviewer 1 Report

Comments and Suggestions for Authors

The Addis Declaration on Immunization  commitment applies to African nations to prioritize immunization as a public health  priority. It was endorsed to ensure that everyone in Africa, regardless of their location or socio-economic status, has access to life-saving vaccines. In this paper oonly assessment to DP and MCV are mentiones. It would be desirable to carry out the same evaluation for polio  or meningitis vaccines for example, although through this declaration, African leaders reaffirm their commitment to eradicating these vaccine-preventable diseases.

In all the work is well wrtitten and presented in orderly manner . In my opinion it is acceptable for publication.

Author Response

Comments 1: The Addis Declaration on Immunization  commitment applies to African nations to prioritize immunization as a public health  priority. It was endorsed to ensure that everyone in Africa, regardless of their location or socio-economic status, has access to life-saving vaccines. In this paper oonly assessment to DP and MCV are mentiones. It would be desirable to carry out the same evaluation for polio  or meningitis vaccines for example, although through this declaration, African leaders reaffirm their commitment to eradicating these vaccine-preventable diseases.

Response 1: Thank you for pointing the need for including polio or meningitis in this analysis in addition to DTP and MCV.  Indeed, this paper analyzed the achievement of the fourth ADI commitment whose indicators are related to MCV and DTP coverage at national and regional level, and health workforce density. Indicators related to vaccine-preventable-diseases elimination including polio are part of the sixth commitment. We are working on a separate paper assessing the achievement of the sixth commitment.  

Comments 2: In all the work is well wrtitten and presented in orderly manner . In my opinion it is acceptable for publication.

Response 2: Thank you for this comment which is much appreciated.

4. Response to Comments on the Quality of English Language

Point 1: No comment from the Reviewer

Response 1:  No response is required.

5. Additional clarifications

Response 1:  No additional clarification from the authors

Reviewer 2 Report

Comments and Suggestions for Authors

Summary - I have read the manuscript with interest. The authors describe the achievement of fourth ADI commitment in Africa as of 2023. The manuscript is well written and methodology is sound. I only have minor suggestions for improvements at this time - 

1. Line 167 - Please include "Indicator 4" at the beginning of the sentence as has been done for other indicators to maintain consistency.

2. Line 169-171  - It would be desirable to provide more details on the distribution of this indicator - specifically the lowest - highest healthcare worker density and the mode of the distribution ( if available). 

3. Line 181 - Consider changing to "Table 3" to stay consistent with the style used elsewhere in the manuscript.

4. Line 190 - Since the acronym has been expanded at the beginning of the manuscript, AUC can be used without expansion here. 

5. Line 194-202—This information is highly relevant to the study's conceptual framework and should thus be included in its background. 

Author Response

Comments 1: Summary - I have read the manuscript with interest. The authors describe the achievement of fourth ADI commitment in Africa as of 2023. The manuscript is well written and methodology is sound. I only have minor suggestions for improvements at this time - 

Response 1: We thank the reviewer for this positive appreciation of our work.

Comments 2: Line 167 - Please include "Indicator 4" at the beginning of the sentence as has been done for other indicators to maintain consistency.

Response 2: Thanks for the suggestion. We have added “Indicator 4” at the beginning of the sentence as suggested (see line 170).

Comments 3: Line 169-171 - It would be desirable to provide more details on the distribution of this indicator - specifically the lowest - highest healthcare worker density and the mode of the distribution ( if available). .

Response 3: Thanks for the suggestion. We have added the following text in lines 172-173 “The median skilled health workers per 10,000 people at national level was 12.5, ranging from 1.4 to 110.9; the mode was 8.0.”

Comments 4: Line 181 - Consider changing to "Table 3" to stay consistent with the style used elsewhere in the manuscript.

Response 4: Thanks for the suggestion. Addressing a comment from the Reviewer 3, we have used Arabic numerals for all tables.

Comments 5: Line 190 - Since the acronym has been expanded at the beginning of the manuscript, AUC can be used without expansion here. 

Response 5: Thanks for the suggestion. We have been applied the guidance (see lines 195-196).

Comments 6: Line 194-202—This information is highly relevant to the study's conceptual framework and should thus be included in its background. 

Response 6: Thanks for the suggestion. We have moved the section providing for the selection of the fourth indicator to the introduction section (see lines 74-81).

4. Response to Comments on the Quality of English Language

Point 1: No comment from the Reviewer

Response 1:  No response required.

5. Additional clarifications

Response 1:  No additional clarification from the authors

Reviewer 3 Report

Comments and Suggestions for Authors

The article is not only interesting, but also important for understanding the progress of vaccination policies in African countries. This assessment highlights the need for more resources for immunization programs in this context.

However, in order to strengthen the presentation, I make the following suggestions:

1. The summary has space to include a few words. For example, explain acronym MCV1: measles first dose (MCV1). Include the % of the 7 states that achieved 3 of the 4 indicators (Same in results, line 173).

2. Suggest moving the first part of the methods to the introduction, including table 1, lines 78 to 80. Refer to Table 1 on line 50, where it talks about the ADI indicators.

3. At the end of the introduction, lines 72-73 repeat the previous paragraph. I suggest merging it into one section. Example: Following the lifting of the public health emergency of international concern (PHEIC) status for the COVID-19 pandemic by WHO on 5 May 2023 [7], assessing the achievement of the ADI commitments is critical.  This paper summarizes progress made by African countries in achieving the indicator related to “Increasing the effectiveness and efficiency of immunization delivery systems as an integrated part of strong and sustainable primary health care systems” by African countries, as of the end 2023.

4. I recommend following the format of the journal in terms of numbering the tables with Arabic numerals (Table 1, 2, 3...), both in the text and in the title of the tables (lines 80, 98, and 108)

5. Start the methods with the presentation of the fourth commitment and table 2.

6. Suggest improving table two by changing ‘Indicator’ to ‘district level indicator’. Example: dropout rate between DTP1 - DTP3. Add a ‘target’ column. Example: Less than 10% per district. Leave “criteria of achievement…” and “data source” the same.

7. I suggest inverting Figure 1, leaving the list of countries on the right and in alphabetical order (as in Figures 2 and 3) and showing the % horizontally. This makes the figures' format uniform and makes them easier to read.

8. Finally, the conclusions mention the need to increase human resources or employ strategies such as training volunteers (community health workers) and improving outreach to users, such as text message reminders. The latter strategy may even be discussed further in the discussion.

Author Response

Comments 1: The summary has space to include a few words. For example, explain acronym MCV1: measles first dose (MCV1). Include the % of the 7 states that achieved 3 of the 4 indicators (Same in results, line 173).

Response 1: Thank you for this suggestion which has been implemented in the summary section (see lines 19-26).

Comments 2: Suggest moving the first part of the methods to the introduction, including table 1, lines 78 to 80. Refer to Table 1 on line 50, where it talks about the ADI indicators.

Response 2: Thank you for this relevant suggestion which we have addressed (see lines 54-58).

Comments 3: At the end of the introduction, lines 72-73 repeat the previous paragraph. I suggest merging it into one section. Example: Following the lifting of the public health emergency of international concern (PHEIC) status for the COVID-19 pandemic by WHO on 5 May 2023 [7], assessing the achievement of the ADI commitments is critical.  This paper summarizes progress made by African countries in achieving the indicator related to “Increasing the effectiveness and efficiency of immunization delivery systems as an integrated part of strong and sustainable primary health care systems” by African countries, as of the end 2023.

Response 3: Thank you for this relevant suggestion. We have changed the last section of the introduction, accordingly, taking also into accounts comments made by the Reviewer 2.  (see lines 79-92).

Comments 4: I recommend following the format of the journal in terms of numbering the tables with Arabic numerals (Table 1, 2, 3...), both in the text and in the title of the tables (lines 80, 98, and 108)

Response 4: Thank you for this relevant suggestion which we have applied throughout the manuscript.

Comments 5: Start the methods with the presentation of the fourth commitment and table 2.

Response 5: Thank you for this suggestion. We have moved the presentation of the fourth commitment to the beginning of the methods section as suggested (see line 104-113)

Comment 6: Suggest improving table two by changing ‘Indicator’ to ‘district level indicator’. Example: dropout rate between DTP1 - DTP3. Add a ‘target’ column. Example: Less than 10% per district. Leave “criteria of achievement…” and “data source” the same.

Response 6: Thank you for this suggestion. We have kept the ADI indicators as worded in the ADI accountability framework. We have added country-level indicators instead of district-level ones and added a column on "target" that corresponds to the criteria for the achievement of the indicator.

Comments 7: I suggest inverting Figure 1, leaving the list of countries on the right and in alphabetical order (as in Figures 2 and 3) and showing the % horizontally. This makes the figures' format uniform and makes them easier to read.

Response 7: Thanks for this suggestion. We have updated Figure 1 as suggested. However, we have kept countries’ names in descending order of the % of districts with less than 10% of DTP1-DTP3 dropout rate.

Comments 8: Finally, the conclusions mention the need to increase human resources or employ strategies such as training volunteers (community health workers) and improving outreach to users, such as text message reminders. The latter strategy may even be discussed further in the discussion.

Response 8: Thank you for this suggestion. Text messing reminders were already discussed as of the strategy of reducing dropout rate (see lines 241-243). We have added a paragraph a reference on the need for investing more in human resources for health in the discussion section (see lines 276-280).

4. Response to Comments on the Quality of English Language

Point 1: No comment from the Reviewer

Response 1:  No response required.

5. Additional clarifications

Response 1:  No additional clarification from the authors